Impact of V-box insertion on promoter activity and virus-inducibility in transgenic Arabidopsis thaliana

Huang Zhenchi 1
Cai Peiyi 1
Huang Meishi 1
Chen Jiating 1
Gan Weixin 1
Wu Limin 1
Li Xiaoming 2
Wu Zhihua wzhua2889@163.com 2 3
1 Life Science & Technology School, Lingnan Normal University , Zhanjiang , Guangdong , China
2 Research Institute of Fast-growing Trees, Chinese Academy of Forestry , Zhanjiang , Guangdong , China
3 State Key Laboratory of Efficient Production of Forest Resources , Beijing , China
Gawande Suresh
Electronic publication date: 2025 Oct 31
Publication date: 2025
Volume: 13
Electronic Location ID: e20178
Received 2025 Apr 16; Accepted 2025 Sep 12
Copyright: ©2025 Huang et al.
Copyright year: 2025
Copyright holder: Huang et al.
License: This is an open access article distributed under the terms of the Creative Commons Attribution License, which permits unrestricted use, distribution, reproduction and adaptation in any medium and for any purpose provided that it is properly attributed. For attribution, the original author(s), title, publication source (PeerJ) and either DOI or URL of the article must be cited.
License URL: https://creativecommons.org/licenses/by/4.0/

Keywords: Synthetic pathogen-inducible promoter, V-box, Transcriptional expression level, Tobacco mosaic virus

Funding: The Natural Science Foundation of Guangdong Province, China No. 2024A1515010649 The Project of Special Fund for Science and Technology Development in Zhanjiang, China No. 2022A01055 This research was funded by the Natural Science Foundation of Guangdong Province, China (No. 2024A1515010649) and the Project of Special Fund for Science and Technology Development in Zhanjiang, China (No.2022A01055). The funders had no role in study design, data collection and analysis, decision to publish, or preparation of the manuscript.

==============================
Synthetic pathogen-inducible promoters (SPIPs) are essential for precise gene regulation in plant genetic engineering. However, natural promoters often exhibit limitations in expression strength and specificity. In this study, we modified the WGFS promoter by incorporating V-box dimers, elements known for their virus-inducible activities, at different positions within its sequence. We thus created three new SPIPs: VWGFS, WGVFS, and WGFSV. These modified promoters were then used in transgenic Arabidopsis thaliana to evaluate their transcriptional properties through β-glucuronidase gene (GUS) staining and Quantitative real-time PCR (qPCR) analysis. Results showed that the insertion of V-box elements at different positions significantly affected the basal transcriptional activity and virus-inducibility. Notably, WGVFS and WGFSV exhibited higher basal activity than VWGFS, with WGFSV showing the highest response to tobacco mosaic virus (TMV) induction. qPCR analysis further revealed that WGFSV had enhanced inducibility by various inducers such as TMV, abscisic acid (ABA), salicylic acid (SA), and ethylene (Eth). This suggests that V-box insertion can alter promoter activity and inducibility, with the midstream position yielding the most desirable characteristics. The transcriptional activities of WGVFS in response to TMV, ABA, SA, and Eth were 1.473, 1.109, 2.030, and 1.088, respectively. Additionally, a typical binary function relationship was observed between the V-box insertion position and transcriptional expression level. The maximum model-predicted value was 1.096 when the V-box was inserted at the 99 bp downstream position. These findings help optimize promoters for plant disease resistance gene control, which may have uses beyond viral induction. This study contributes to the development of synthetic promoters with specific activity for plant genetic engineering.

Introduction

In the genetic engineering of plant disease resistance, precise spatial and temporal control of transgene expression is crucial to reduce potential negative impacts on plant growth and yield (Gurr & Rushton, 2005). Constitutive expression of defense genes can lead to poor plant quality (Gurr & Rushton, 2005), so strategies that limit expression to infection sites or specific times are preferred.

Promoters are DNA sequences located upstream of the 5′ end of structural genes. They are critical for initiating transcription and the regulation of gene expression in plants (Dutt et al., 2014; Villao-Uzho et al., 2023). They are recognized and bound by RNA polymerase, serving as transcription start sites. These cis-acting elements regulate gene expression by modulating its level, site, and mode (Huang et al., 2019). However, natural promoters often exhibit limitations in terms of expression strength and specificity (Aysha et al., 2018). To address these challenges, an ideal synthetic pathogen-inducible promoter has been developed and engineered to regulate gene expression spatially and temporally. Inducible and tissue-specific promoters are preferred over constitutive ones to minimize adverse effects on non-target organisms (Kummari, PS & Kishor, 2020; Villao-Uzho et al., 2023). SPIPs exhibit several desirable properties: broad responsiveness to inducers, rapid initiation of expression, high induction efficiency, minimal basal activity, and damage independent induction (Huang et al., 2017). These characteristics allow SPIPs to provide precise and tunable control of gene expression, enabling responses to varying inducer levels for fine regulation of gene activity (Huang et al., 2017; Rohlhill, Sandoval & Papoutsakis, 2017; Baldin et al., 2022), providing significant potential for improving crop productivity and promoting sustainable agricultural development (Ali & Kim, 2019).

Multiple cis-regulatory elements are located upstream of the 5′ region of plant genes, where their distribution and presence influence the gene’s expression pattern (Dutt et al., 2014). In the context of designing synthetic/artificial promoters, the arrangement of these cis-acting elements plays a crucial role in determining the transcriptional properties of the gene (Metzger et al., 2016; Jores et al., 2021). Consequently, selecting appropriate cis-acting elements and optimizing their positional relationships can facilitate the development of ideal synthetic promoters, enabling the spatiotemporal regulation of target gene expression in genetic engineering. Initially, synthetic promoter research focused on three strategies: combining cis-regulatory elements with strong constitutive promoters, integrating cis-regulatory elements from different promoters, or fusing two constitutive promoters to create bidirectional promoters (Mehrotra et al., 2011). Various tetrameric synthetic promoters have been developed using elements such as the W1-box, revealing significant differences in basal expression levels, inducible factors, response speeds, and transcriptional activities among promoters (Rushton et al., 2002). Venter reviewed advancements in artificial promoter design, emphasizing the attachment of specific cis-regulatory elements individually or in combination with minimal constitutive promoters to generate highly active constructs (Venter, 2007). Subsequent studies demonstrated that optimizing natural promoter elements and generating synthetic promoters with desirable characteristics can lead to highly effective promoters in plants (Liu & Stewart, 2016). An ideal pathogen-inducible promoter can be constructed by combining elements (Peng et al., 2011), such as F-box (5′-TTGTCAATGTCATTAAATTCAAACATTCAACGGTCAATT-3′) (Heise et al., 2002), S-box (5′-CAGCCACCAAAGAGGACCCAGAAT-3′) (Kirsch et al., 2000), Gst1-box (5′-TTCTAGCCACCAGATTTGACCAAAC-3′) (Malnoy et al., 2006), and W-box (5′-TTATTCAGCCATCAAAAGTTGACCAATAAT-3′) (Wang et al., 1998). Huang et al. (2017) designed eight artificial promoters—FSGW, FSWG, GWFS, GWSF, SFGW, SFWG, WGFS, and WGSF—by incorporating Gst1-box, W-box, S-box, and F-box elements into the minimal CaMV35S promoter (−46 to +8, TATA box). Their findings highlighted that variations in the positions of cis-acting elements significantly affected transcriptional properties (Peng et al., 2011; Huang et al., 2017). Among these, GWSF and WGFS displayed the most desirable induction characteristics.

The WGFS promoter exhibits low basal expression in transgenic Arabidopsis thaliana and can be induced by Ralstonia solanacearum, Phytophthora capsici, and SA (Huang et al., 2017). Beyond bacterial and fungal pathogens, viruses represent another significant class of plant pathogens, causing substantial harm. Introducing a cis-acting element with virus-inducing activity into the WGFS promoter can enhance its functionality, enabling it to respond to bacterial, fungal, and viral inductions. However, key questions remain unanswered: Does the position of the V-box inserted element correlate with viral induction activity? Does the modified promoter retain its original inducible activities after element insertion? These questions are rarely addressed in current literature.

In this study, we optimized the WGFS promoter by incorporating V-box dimers, known for their virus-inducible activity (Kobayashi et al., 2010), at different positions within the promoter sequence. Specifically, V-box dimers were inserted upstream, midstream, or downstream of the WGFS sequence, resulting in three modified promoters: VWGFS, WGVFS, and WGFSV. The transcriptional properties of these modified promoters were assessed in transgenic A. thaliana through β-glucuronidase gene (GUS) histochemical staining and qPCR analysis. The assessments compared the effects of V-box positioning on the transcriptional activity of WGFS and GWSF. This work provides valuable insights into the design and optimization of synthetic promoters. The findings are expected to guide the development of ideal promoters for plant disease resistance genetic engineering.

Materials & Methods

Materials

In this study, A. thaliana Columbia ecotype (Col-0) was used as the experimental material. Arabidopsis plants were cultivated in vermiculite under controlled environmental conditions in a climatic chamber. The specific growth parameters were as follows: a light intensity of 300 µmol m−2 s−1, a 12-hour photoperiod at 26 °C, followed by a 12-hour dark period at 20 °C, with relative humidity maintained at 70–80%. The Agrobacterium tumefaciens strain GV3101 was used for genetic transformation in this experiment.

Improved synthetic promoter design and vector construction

In this study, two V-box sequences (TTGGGAAGGAATTTCCTACT) were ligated to form a V-box dimer by introducing a six bp ACTAGA sequence as a spacer. The V-box dimer was inserted into the upstream, midstream, and downstream regions of the WGFS promoter using a 10 bp DNA linker (GAAGATAATC). The insertion process yielded three modified promoters: VWGFS, WGVFS, and WGFSV (Fig. 1). To facilitate cloning, HindIII and BamHI restriction sites were incorporated into the upstream and downstream regions of the modified promoters, respectively. All sequences were synthesized by Sangon Biotechnology (Shanghai, China) and cloned into the pUC19 vector.

Figure 1 A schematic diagram of WGFS, VWGFS, WGVFS and WGFSV SPIPs.

G, Gst1-box; W, W-box; S, S-box; F, F-box; V, V-box; IS, six bp DNA insert sequence (ACTAGA); CS, 10 bp DNA connection sequence (GAAGATAATC); Mini 35S, the minimal CaMV35S promoter.

The CaMV35S promoter fragment in the pBI121 vector was replaced with each modified promoter to regulate the expression of the gus gene. This was achieved through restriction enzyme digestion and ligation. The recombinant plasmids were then introduced into Agrobacterium tumefaciens strain GV3101 using the freeze-thaw transformation method. The successful construction of the vectors and Agrobacterium transformation were confirmed through PCR analysis, enzymatic digestion, and sequencing.

The primers used in the PCR assay were as follows: For the VWGFS and WGVFS promoters, the primers were identical: Forward: 5′-TTTCCTACTACTAGATTGGGAAGGA-3′/Reverse: 5′-GGAAGGGTCTTGCGGATTAT-3′. The amplified fragment length was 379 bp for VWGFS and 237 bp for WGVFS. For WGFSV, the primer sequences were: Forward: 5′-TGAAGATAATCCAGCCACCA-3′/ Reverse: 5′-AGCGTGTCCTCTCCAAATGA-3′. The amplified fragment length was 205 bp. For NPTII, the primer sequences were: Forward: 5′-GAGGCTATTCGGGCTATGACTG -3′/ Reverse: 5′-ATCGGGAGCGGCGATACCGTA-3′. For the gus gene, the primer sequences were: Forward: 5′-ACACCGATACCATCAGCG-3′/Reverse (R): 5′-TCACCGAAGTTCATGCCAT-3′.

Genetic transformation and screening

Single colonies of the Agrobacterium tumefaciens GV3101 engineering strain were isolated and cultured using the plate streaking method to activate the bacteria. Genetic transformation of wild-type A. thaliana was then performed using the Agrobacterium-mediated floral dip method (Zhang et al., 2006). Seeds from the transformed plants were collected upon maturation.

The harvested seeds were germinated on 1/2 MS solid medium supplemented with 50 mg L−1 kanamycin to screen for potential transformants. For each modified promoter, over 30 independent T0 generation plants were selected. Five-week-old transgenic seedlings were further analyzed by PCR to confirm the presence of the SPIP, NPTII, and gus genes.

To identify transgenic plants harboring a single copy of the T-DNA insert, over 3,000 T1 generation progeny derived from each T0 plant were screened for kanamycin resistance. The green-to-yellow seedling ratio was determined, and plants with about a 3: 1 segregation ratio were identified as single-copy transgenic lines. Subsequently, five single-copy T1 plants were cultivated, and the same screening method was applied through successive generations up to T3 to establish stable transgenic lines. Seeds from T3 generation lines expressing CaMV35S: gus or WGFS: gus were preserved for further use in the laboratory (Huang et al., 2017).

Induction treatment and GUS staining

Five-week-old T3 generation A. thaliana plants, as described above, were used for GUS staining to evaluate the basal expression activity of the modified promoters, following the method of Jefferson, Kavanagh & Bevan (1987). Induction treatments were carried out under various stress conditions, including high temperature, low temperature, high osmotic stress, and viral infection with tobacco mosaic virus (TMV). The specific treatment conditions were as follows:

High temperature treatment: Plants were placed in an artificial climatic chamber that was set to 37 °C for 12 h.

Low temperature treatment: Plants were subjected to a temperature of 4 °C for 12 h in the climatic chamber.

High osmotic treatment: Plants were grown in 100 mL of vermiculite and irrigated with 50 mL of a 200 mmol L−1 NaCl solution for 12 h.

TMV treatment: Virus particles were extracted from severely symptomatic Nicotiana tabacum L. cv. Xiangyan No. 3 leaves. The leaves were ground in three times the volume of 30 mmol L−1 phosphate buffer (pH 7.2), and the resulting homogenate was centrifuged at 4 °C and 3,000 g for 5 min. Chloroform was added (1/5 the volume of the leaves) to remove proteins, and the supernatant was centrifuged at 4 °C and 3,000 g for 10 min. Polyethylene glycol (PEG6000) was added to the supernatant to achieve a 6% (m/v) concentration, followed by NaCl to a final concentration of 0.5 mol L−1. Following 12 h of precipitation, the precipitate was collected by centrifugation at 4 °C and 3,000 g for 30 min. The precipitate was resuspended in 0.9% (m/v) NaCl, and the supernatant containing TMV particles was used as the inducer after centrifugation at 4 °C and 3,000 g for 30 min. Rosette leaves of five-week-old seedlings were inoculated with 50 µL TMV particle solution.

Abiscisic acid (ABA) and salicylic acid (SA) induction treatments: The rosette leaves were sprayed with two mmol L−1 ABA or 75 µmol L−1 SA solutions, respectively. All treatments were conducted for 12 h.

Ethylene (Eth) treatment: This was performed according to the method described in our previous publication (Huang & Li, 2018). The ethylene treatment was performed by placing Arabidopsis seedlings in sealed plastic containers along with 200 mL conical flasks containing 1% (m/v) NaHCO3. Following the release of ethylene, an appropriate volume of ethephon solution was added to the flasks to achieve a final concentration of 0.40 mmol L−1. Arabidopsis plants were exposed to ethylene treatment for 12 h in sealed containers.

Following the induction treatments, GUS staining was performed, and the plants were subsequently destained in alcohol. Photographs were taken for analysis, and the resulting images were used to assess the transcriptional activity of the promoters in the transgenic plants.

qPCR assay

Plant materials were prepared in triplicate for each treatment. Fifteen A. thaliana T3 generation plants were used per treatment. After 12 h of inducer treatments, the 15 plants were divided randomly into three groups. From each group, one rosette leaf was collected from five randomly selected plants to form a single biological replicate. The collected leaves were ground in liquid nitrogen, and total RNA was extracted using the UNIQ-10 column and Trizol total RNA isolation kit (Sangon Biotech, Beijing, China). The quality and concentration of the extracted RNA were assessed using a NanoDrop 2000C spectrophotometer (Thermo Fisher, Waltham, MA, USA).

For reverse transcription, total RNA was used as a template following the instructions of the PrimeScript RT kit (TaKaRa, Shiga, Japan). The resulting cDNA served as a template for qPCR. qPCR was performed using the SYBR Premix Ex Taq™ II reagent (Tli RNaseH Plus, TaKaRa, Shiga, Japan) on a CFX96 Real-Time PCR System (Bio-Rad, Hercules, CA, USA). The total reaction volume was 25 µL and contained 12.5 µL of 2 × SYBR Premix Ex Taq™ II. The EF1-α gene (AT5G60390) was selected as the internal reference. The primers used were:

gus primers: Forward: 5′-CTGATAGCGCGTGACAAAAA-3′, Reverse: 5′-GGCACAGCACATCAAAGAGA-3′. Its amplified fragment length was 211 bp.

EF1-α primers: Forward: 5′-TGAGCACGCTCTTCTTGCTTTCA-3′, Reverse: 5′-GGTGGTGGCATCCATCTTGTTACA-3′. Its amplified fragment length was 76 bp.

The qPCR amplification program was as follows:

Pre-denaturation at 94 °C for 1 min,

Denaturation at 94 °C for 13 s,

Annealing at 55 °C for 13 s,

Extension at 72 °C for 15 s,

A total of 40 cycles were performed. Reaction specificity was assessed by melting curve analysis at 65–90 °C.

The transcriptional activity of the CaMV35S promoter in CaMV35S: gus transgenic T3 generation plants was set as the reference (relative value = 1.0); the relative transcriptional activity of the modified promoters was calculated using the relative quantification method (Pfaffl, 2002).

Experimental design and statistics

In the transcriptional activity assay, a three-way analysis of variance (ANOVA) design incorporating promoters from various V-box positions and inducer treatments was employed. Statistical analyses were performed using ANOVA, followed by the Holm-Bonferroni test to identify significant differences between groups. Differences were considered statistically significant at p < 0.05.

Results

Effect of V-box insertion position on promoter basal transcriptional activity

The activity of β-glucuronidase (GUS) was detected through histochemical staining. The blue intensity reflects promoter activity strength. Wild-type (WT) A. thaliana plants served as the negative control, while CaMV35S: gus T3 generation plants were used as the positive control. GUS staining showed clear differences in the effect of V-box insertion positions on the basal transcriptional activity of the WGFS promoter (Fig. 2). The staining intensity of VWGFS, in which the V-box was inserted upstream, was similar to that of the original WGFS promoter, indicating maintained low basal transcriptional activity. In contrast, the midstream and downstream insertion of the V-box noticeably enhanced the staining intensity for the WGVFS and WGFSV promoters, indicating a substantial increase in their basal transcriptional activity compared to the original WGFS promoter.

Despite these differences, the native transcriptional activities of all modified promoters were substantially lower than that of the positive control CaMV35S promoter. These findings indicate that V-box insertion alters the basal transcriptional activity of the promoter, and that the insertion position has a pronounced effect on this activity.

Response of the modified promoter to low temperature, high temperature, high osmotic, and TMV treatment

An ideal pathogen-inducible promoter should exhibit a broad range of pathogen-inducible activity while remaining unresponsive to non-pathogen factors. In this study, GUS staining was used to assess the transcriptional activity of the modified promoters following treatments with low temperature, high temperature, hyperosmolarity, and TMV. The results are presented in Fig. 3. The basal expression level of VWGFS was relatively low, with minimal changes in transcriptional activity following TMV treatment. Transcriptional levels were lower under low-temperature conditions but increased significantly after high temperature and high osmotic treatments. When the V-box was inserted upstream, the transcriptional properties of VWGFS closely resembled those of the original WGFS promoter. However, when the V-box was inserted into the midstream or downstream positions, the basal transcriptional levels of WGVFS and WGFSV were significantly higher.

Figure 2 The basal transcriptional levels of the improved promoters in GUS transgenic Arabidopsis thaliana by GUS staining evaluation.

Each panel shows representative GUS-stained leaves from independent T3 transgenic lines for each promoter construct. The images illustrate biological replicates from separate transformation events.

Figure 3 The expression levels in leaves of gus transgenic Arabidopsis thaliana with improved promoter under four stresses by GUS staining.

CK, The basal expression level of promoters without stress inducer treatment. Low temperature stress treatment: Place the plants in a climate chamber at 4 °C for 12 h. High temperature stress treatment: Plants were placed in an artificial climate chamber at 37 °C for 12 h. High osmotic treatment: Plants were irrigated with 50 mL of 200 mmol L−1 NaCl solution for 12 h. Tobacco Mosaic Virus (TMV) stress treatment: Rosette leaves of 5-week-old Arabidopsis thaliana were inoculated with 50 µL TMV virus particle solution and treated for 12 h.

Both WGVFS and WGFSV exhibited lower transcriptional levels under low temperature treatment, while transcriptional levels after high temperature and high osmotic treatments were comparable to basal levels, with no significant changes in staining intensity. Upon TMV induction, the transcriptional levels of WGVFS and WGFSV were elevated, with the increase being more pronounced in WGFSV. These results suggest that the insertion position of the V-box influences the inducibility of the modified promoter in response to TMV, with stronger induction observed when the V-box is positioned closer to the minimal CaMV35S sequence.

qPCR assessment of modified promoter transcriptional properties

GUS staining provides visual evidence of promoter transcriptional activity; however, it is not precise for quantifying transcriptional levels. To achieve accurate quantification, qPCR was used to measure the transcriptional levels of the gus reporter gene. The transcriptional activity of the CaMV35S promoter in CaMV35S: gus T3 generation plants was set as the reference value of 1.0. The transcriptional activities of the synthetic promoters were then quantified by calculating the relative expression levels of gus gene using the Pfaffl relative quantification method (Fig. 4).

Figure 4 Transcriptional expression levels of improved promoters in transgenic Arabidopsis thaliana after treatment with different inducers.

Basal refers to the basic activity of promoters in the absence of stress treatment. TMV represents the expression level of promoters following treatment with tobacco mosaic virus. ABA indicates the expression level of promoters after ABA treatment. Similarly, SA denotes the expression level of promoters in response to salicylic acid treatment, while Eth represents the expression level of promoters following ethylene treatment. Five-week-old Arabidopsis thaliana rosette leaves were inoculated with 50 µL of TMV virus pellet solution. For ABA and SA induction treatments, the leaves were sprayed with two mmol/L ABA and 75 µmol L−1 SA solutions, respectively. All treatments were conducted for 12 h. The ethylene treatment was performed by placing Arabidopsis seedlings in sealed plastic containers along with 200 mL conical flasks containing 1% (m/V) NaHCO3. Following the release of ethylene, an appropriate volume of ethephon solution was added to the flasks to achieve a final concentration of 0.40 mmol L−1. Seedlings were exposed to ethylene treatment for 12 h in the sealed containers. Significant differences among promoters under identical treatments are indicated by lowercase letters of the same color on each bar, with significance determined at the 0.05 level. Statistical analyses were performed using ANOVA (analysis of variance), followed by the Holm-Bonferroni test to identify significant differences between groups.

qPCR revealed that the WGFS promoter had a basal activity of 0.160. After inducer treatments, the relative transcriptional activities were as follows: 0.185 for TMV, 0.532 for ABA, 0.392 for SA, and 0.777 for Eth. WGFS was significantly induced by ABA, SA, and Eth, with transcriptional levels elevated more than two folds. VWGFS exhibited a lower basal transcriptional activity of 0.048, with relative transcriptional activities after induction by TMV, ABA, SA, or Eth of 0.157, 0.062, 0.243, and 0.328, respectively. Although VWGFS was responsive to all inducers, its transcriptional activity remained relatively low compared to the other promoters.

WGVFS showed a basal activity of 0.877. After TMV, ABA, SA, or Eth induction, transcriptional activities increased to 1.473, 1.109, 2.030, and 1.088, respectively, SA produced the strongest response. WGFSV exhibited a basal transcriptional activity of 0.600, which increased to 1.875, 0.741, 0.764, and 1.203 after the treatments with TMV, ABA, SA, or Eth, respectively. TMV induction elevated WGFSV transcriptional activity by more than threefold, while Eth also induced a more than twofold increase in transcriptional activity.

A comparison of GUS staining and qPCR results revealed nearly identical trends and results, indicating that V-box insertion significantly enhanced the transcriptional activities of the modified promoters, particularly after TMV induction, with transcriptional activities more than twice their basal levels. These results indicate that V-box insertion confers TMV-induced transcriptional activity, with the level of induction strongly influenced by the V-box position within the promoter. Promoters with V-box closer to the minimal CaMV35S exhibited higher transcriptional activity following TMV induction. Additionally, the modified promoters retained their original inducible properties in response to ABA, SA, and Eth, indicating that V-box insertion did not compromise the specific inducibility of the original promoter.

To further investigate the effect of V-box position on the transcriptional properties of different promoters, a three-way ANOVA as implemented to compare the responses of the WGFS promoter and the GWSF promoter to various inducers. The factors included two promoters (Factor A: Promoter), four V-box positions (Factor B: Position), and five inducer treatments (Factor C: Inducer).

Each treatment combination was repeated three times, and relative expression levels (log10-transformed) were analyzed using ANOVA to evaluate all main effects and interaction effects. The results presented in Table 1 indicate that the model F-value of 40.58 demonstrates statistical significance, with a coefficient of determination (R2) of 0.995 (Table S1) and a p-value < 0.0001. In the model, the p-values of A, B, C, A×B, A×C, B×C, and A×B×C are all less than 0.001, indicating significant main and interaction effects. The F-value ranking of the factors, from highest to lowest, is Factor B (2019.23) >Factor C (498.73) >Factor A (165.98).

Table 1 ANOVA for selected factorial model analysis of variance table.

Source	Sum of squares	df	Mean square	F-value	P-value	
Model	28.42	39	0.73	408.58	<0.0001	
A-Promoter	0.30	1	0.30	165.98	<0.0001	
B-V-box position	10.80	3	3.60	2,019.23	<0.0001	
C-Inudcers	3.56	4	0.89	498.73	<0.0001	
A×B	1.48	3	0.49	276.55	<0.0001	
A×C	0.36	4	0.09	50.85	<0.0001	
B×C	6.54	12	0.54	305.50	<0.0001	
A×B×C	5.38	12	0.45	251.41	<0.0001	
Residual	0.14	80	0.00			
Corrected total	28.56	119				

These results demonstrate that the V-box position (Factor B) has the most significant influence on transcriptional activity, followed by inducer treatment (Factor C) and promoter type (Factor A). According to the above model, the transcriptional activities of WGFS and GWSF promoters under various factor combinations are shown in Fig. 5. WGFS and GWSF promoters exhibit negligible transcriptional levels (close to 0) in the absence of V-box insertion (CK) or no inducer treatment (CK1), indicating that both V-box insertion and inducer stimulation are essential for promoter-driven expression. Additionally, the WGFS and GWSF promoters demonstrate significant differences in their response to the same inducer treatments. Under no inducer treatment conditions (CK1), the GWSF promoter displayed a slightly higher basal expression level than the WGFS promoter. However, both promoters were particularly sensitive to Eth treatment, which significantly increased expression levels. Notably, the relative expression level of the GWSF promoter under Eth treatment was approximately double that of the WGFS promoter. Compared to CK1, the relative expression levels of both WGFS and GWSF promoters under TMV induction showed little variation (Fig. 5 and Fig. S1). In contrast, the WGVFS promoter demonstrated a marked increase in expression level under TMV induction relative to CK1 (Fig. S1). Meanwhile, the GWVSF promoter exhibited a modest expression level of 0.181 under TMV induction, only slightly higher than its CK1 control value of 0.159 (Fig. S1). These results indicate that the WGVFS promoter has significantly stronger transcriptional activity under TMV induction than the GWVSF promoter (Fig. S1).

Figure 5 Model-based prediction of relative expression levels of the modified GWSF and WGFS promoters under various combinations of factors.

(A) GWSF and modified promoters, derived from different V-box insertion positions in the GWSF promoter. (B) WGFS and modified promoters, resulting from variations in V-box insertion positions within the WGFS promoter; CK indicates the promoters without V-box element insertion. CK1 indicates the promoter transgenic plants not treated with any of the four inducers (ABA, SA, Eth, and TMV). The varying bar heights in the figure represent the relative expression levels of different factor combinations, including the WGFS and GWSF promoters under various inducer treatments, as predicted by a model derived from a three-way ANOVA analysis (Table 1). Transcriptional expression levels are expressed as relative values, calculated as the ratio of the expression level of each promoter to that of the CaMV35S promoter.

For SA and ABA treatments, the modified GWSF promoter displayed slightly higher local expression levels compared to CK(GWSF) promoter, especially at the midstream V-box insertion position. Overall, the WGFS promoter showed a more balanced induction response to TMV, SA, and ABA treatments, while the GWSF promoter exhibited a more specific response, particularly to Eth. Among the four inducers (TMV, SA, ABA, and Eth), Eth was the most potent inducer for both promoters (Fig. 5), followed by TMV. The position of V-box insertion significantly influences transcriptional activity under the same inducer treatment. The upstream insertion position resulted in the lowest expression levels among all inducers. In contrast, the midstream insertion position yielded the highest expression levels for all inducers, demonstrating the strongest positive factor effect on transcriptional activity, particularly in WGFS promoter plants. The downstream insertion position showed intermediate factor effects on transcriptional activity (Fig. 6).

Figure 6 Transcriptional expression level means of four V-box positions in eight promoters.

Each data point in the graph represents the average of relative expression levels (listed in Table S2), and each V-box position has the average of 30 expression levels under five inducer treatments. Colors indicate different promoters in the graph.

Relationship between V-box insertion position and promoter transcriptional activity

We analyzed and compared the effects of different V-box insertion positions on the cis-element positional effects of the two promoters, WGFS and GWSF (Figs. S2–S4). The sequence arrangement of these promoters was aligned as follows: GW (no V-box inserted between the Gst1-box and W-box) >W-box >Gst1-box (Fig. 6). Notably, the positional effects of the V-box were stronger than those of the other three cis-elements, highlighting their critical role in regulating transcriptional activity (Fig. 6). We also analyzed the relationship between the V-box insertion position and transcriptional activity, which exhibited a typical binary function relationship, with the equation: y = 0.93193 + 0.00332 x − 1.674555  × 10−5 x2 (Fig. 7). According to the equation, the maximum predicted expression value of 1.096 was observed when the V-box was inserted at the 99 bp position, corresponding to the downstream location (Fig. 7). The data further indicate that both very close and very distant V-box insertions lead to reduced expression levels. In contrast, an optimal intermediate distance, approximately 99 bp, significantly enhances promoter activity. These findings suggest that the spatial arrangement of cis-elements is crucial for efficient transcriptional activation. The optimal positioning of the V-box likely facilitates the recruitment of transcription factors (TFs) and the proper assembly of the transcriptional machinery, thereby maximizing promoter activity.

Figure 7 Equation between four V-box positions in eight promoters of transgenic Arabidopsis thaliana and their transcriptional expression levels after inducer treatment.

The relative expression levels (Y-axis) were determined by qPCR and are shown as fold changes compared to the CaMV35S promoter. The X-axis represents the distance (in base pairs) from the V-box insertion position to the minimal promoter. A regression analysis was performed, treating the V-box position as the independent variable and the relative expression level as the dependent variable. For the promoter without V-box, there is no insertion position, so the distance to the minimal promoter cannot be defined, in order to eliminate the effect of the promoter without V-box on the promoter activity, so we set the distance of the promoter without V-box from the minimal promoter to x = 0 bp, which can be well able to be used in the binary equation to reflect its x effect. The resulting binary function is y = 0.93193 + 0.00332x − 1.674555 ×10−5×2 with high R-square value (0.99373) indicating an excellent fit between the model and the experimental data. According to the model, the maximum predicted expression occurs when the V-box is inserted 99 bp downstream of the transcription start position, reaching a value of 1.096 times that of the CaMV 35S promoter.

Discussion

Effect of V-box insertion at different positions in the promoter sequence on promoter transcriptional properties

The transcriptional properties of promoters are influenced by numerous factors, including the function and copy number of cis-acting elements, the spacing of the TATA-box, and the position of these elements relative to their regulatory targets (Shokouhifar et al., 2011). This study specifically evaluated the effects of inserting virus-inducible cis-element (V-box) dimers at different positions (upstream, midstream, or downstream) within the synthetic promoter WGFS. Although the structural framework of the three modified promoters was identical (Fig. 1), the position of V-box insertion produced distinct transcriptional outcomes. Upstream V-box insertion in VWGFS significantly reduced basal transcriptional levels compared to WGFS. In contrast, midstream and downstream V-box insertions in WGVFS and WGFSV, respectively, resulted in markedly increased basal activity relative to WGFS (Fig. 2).

The transcriptional process is tightly regulated by sequence-specific DNA-binding proteins, which are known as TFs, that modulate gene activity in specific cell types (De Jonge et al., 2022). Transcriptional patterns are dictated by cis-regulatory elements—DNA sequence motifs that interact with specific TFs (Marand et al., 2023). The positional arrangement of DNA in the nucleus further influences the interaction between TFs and promoters. Despite WGFS and GWSF promoters containing identical cis-elements, differences in their basal activities affect their transcriptional responses to various inducers (Fig. 5). This suggests that the V-box insertion position may either create or disrupt binding sites for specific TFs, altering transcriptional activity. Alternatively, V-box insertion may influence nucleosome positioning, affecting TF accessibility to DNA (Marand et al., 2023). These changes can modulate TF binding efficiency, and overall promoter activity.

Additionally, changes in DNA spatial conformation caused by V-box insertion can alter histone modifications and chromatin accessibility, which are critical for transcriptional regulation.

Such changes can impact TF binding efficiency, RNA polymerase II recruitment, and transcription initiation. Thus, varying V-box insertion positions can significantly influence basal transcriptional levels by affecting chromatin structure, TF interactions, and transcriptional machinery accessibility.

In plants, transcriptional regulatory sequences are abundant, and their positional context is crucial for transcriptional regulation. Regulatory elements located downstream of the transcription start site (TSS) play a central role in transcriptional expression (Voichek et al., 2024). Altering the positional relationships between plant regulatory elements and the TSS has a profound impact on transcriptional activity. Our findings revealed that upstream insertion of the V-box in VWGFS lowered basal transcriptional levels below those of WGFS. This reduction is likely attributed to an unfavorable spatial distance between the V-box and the core promoter region, which may facilitate binding of repressive TFs. By contrast, midstream and downstream V-box insertions in WGVFS and WGFSV significantly increased basal activity (Fig. 2), likely by altering changes in promoter spatial conformation to improve recruitment efficiency of RNA polymerase II and transcription factor recruitment efficiency.

In the WGFS and GWSF promoters, V-box insertion positions had a more significant impact on transcriptional activity compared to other three cis-elements (Fig. 6). A binary functional relationship was observed between V-box insertion position and promoter transcriptional activity. Specifically, when the V-box was inserted at the 99th bp, the model predicted a maximum expression value of 1.096 (Fig. 7). These results are consistent with previous studies that have demonstrated that alterations in the positional context of cis-regulatory elements significantly influence transcriptional properties (Kobayashi et al., 2010; Huang et al., 2022; Voichek et al., 2024). This study highlights the important role of spatial configuration in modulating transcriptional activity and provides insights for optimizing promoter design to enhance gene regulation in synthetic biology and biotechnology applications.

V-box insertion affects promoter responsiveness to virus induction

The V-box is a virus-inducible cis-element that plays a pivotal role in regulating promoter activity under a variety of stimuli. In this study, TMV, ABA, SA, and Eth treatments significantly increased the expression levels of WGVFS and WGFSV compared to WGFS. Among the three promoters, WGFSV exhibited the highest expression levels, particularly under TMV and Eth induction. The proximity of the V-box to the minimal CaMV35S core promoter strongly influenced expression activity, with closer positioning resulting in more pronounced changes and higher inducibility across multiple factors (Figs. 3 and 4).

Notable differences among the promoters were observed. The WGFS promoter exhibited greater sensitivity to TMV and SA induction, making it particularly suitable for studies related to virus response. Significant interactions between the A×B and B×C terms (Table 1) revealed that the impact of the V-box insertion position (Factor B) varied across promoters under different inducer treatments. Overall, downstream V-box position positively influenced TMV and SA responses but negatively affected Eth-induced activity. The magnitude of these effects was promoter- and inducer-specific: the WGFS promoter displayed high sensitivity to midstream V-box insertion, while the GWSF promoter showed elevated relative expression levels when the V-box was inserted in the downstream position.

The results demonstrated that V-box dimer insertion significantly altered the promoter transcriptional properties, leading to a marked increase in expression activity under various inducer treatments. Eth- and viral- induced activity was particularly enhanced as the V-box insertion position shifted downstream, likely due to the presence of stress-responsive cis-elements, such as the W-box, within the promoter. These cis-elements bind specific TFs that interact synergistically to enhance promoter activity and responsiveness to viral induction.

Beyond positional effects, the number of V-box elements also plays a critical role in modulating promoter activity. Previous studies in bacteria and yeast demonstrated that increasing the number of promoter or binding site copies could enhance gene expression, although this effect is often influenced by transcription factor availability and chromatin context (Brewster et al., 2014; Hull et al., 2017; Segall-Shapiro, Sontag & Voigt, 2018). For example, in apple, introducing synthetic TATA box repeats into the promoter of the iron transporter gene IRT1 resulted in a positive correlation between promoter activity and gene expression (Zhang et al., 2017). In this study, promoters containing V-box elements exhibited modest increases in basal expression levels, particularly the WGVFS promoter under TMV treatment. This suggests that the V-box copy number influences both basal and induced expression. It is plausible that increasing the number of V-box elements could further enhance TMV inducibility and improve the dynamic range between basal and induced expression. However, this strategy may risk higher basal activity or transcription factor titration effects (Brewster et al., 2014).

Future research should systematically investigate how V-box copy number and position affect promoter induction strength. Optimizing these parameters could lead to the development of synthetic promoters with low basal activity and strong, specific induction in response to viral infection. Such promoters could have broad applications in plant molecular studies and biotechnology, enabling precise control of gene expression under stress conditions.

Modified promoters retain the original promoter response to other inducers

Core promoter elements represent the most distinct class of cis-regulatory modules due to their predictable positioning around the transcription start site (TSS) of a gene (Marand et al., 2023). These elements serve as platforms for assembling the preinitiation complex (PIC), which includes RNA polymerase II and general TFs. This assembly is facilitated by the normally accessible chromatin conformation maintained by the unstable histone variant H2A.Z (Hampsey, 1998; Haberle & Stark, 2018). Typically, the assembly of this complex is sufficient to support basal transcription.

Our experimental results demonstrated that the responsiveness of each modified promoter to other inducible factors, such as ABA, SA, and Eth, was not compromised by the introduction of the V-box. This suggests that the addition of the new cis-regulatory element did not disrupt the local chromatin environment of the original promoter. Instead, the V-box enhanced the promoter’s responsiveness to viral induction, broadening its range of inducibility. In this study, we successfully designed modified synthetic promoters with enhanced responsiveness to pathogen-related inducers by strategically inserting V-box elements at different positions. These findings provide a valuable reference for applications in disease resistance genetic engineering.

To further evaluate the specificity of these modified promoters, it is essential to examine their behavior under diverse conditions. While our study did not quantitatively measure promoter activity under abiotic stress treatments like low temperature, high temperature, and high osmotic stress using qPCR, we did rely on GUS staining for a qualitative assessment. Incorporating qPCR data for these abiotic stressors in future work will provide a more precise evaluation of promoter specificity. Future research will include qPCR analysis under various abiotic stresses to confirm the modified promoters’ specific responsiveness to pathogen-related stimuli, ensuring their targeted utility for genetic engineering.

Although our results show that the modified promoters retain inducibility by multiple chemical inducers and respond strongly to TMV, we did not directly test their response to other classes of pathogens (such as bacteria or fungi) in this study. Future studies should evaluate the activities of these promoters in response to both bacterial and fungal pathogen challenges to comprehensively determine their effectiveness as broad-spectrum, pathogen-inducible promoters. Additionally, further research should expand these investigations to include a wider variety of plant models and diverse environmental conditions. This approach will help validate the robustness and versatility of the synthetic promoters across different biological contexts.

Conclusions

In this study, we used V-box insertion to enhance pathogen-inducible promoters in A. thaliana. Compared with the strong CaMV35S promoter, WGVFS and WGFSV showed significantly higher activities under all tested inducers. Notably, WGVFS demonstrated efficient induction by ABA and SA. Compared to the GWSF promoter, the WGFS promoter exhibited strong inducibility by TMV and Eth, while retaining efficient responsiveness to SA and ABA. These findings provide valuable insights into optimizing promoter sequences to improve responsiveness to pathogen-related inducers. The development of synthetic promoters with enhanced specificity and inducibility offers a promising strategy for applications in genetic engineering aimed at improving disease resistance in plants.

Supplemental Information

Supplemental Information 1 The Model fit goodness and prediction

Std. Dev.: Standard deviation; b=C.V. %: Coefficient of variation percentage; PRESS: Predicted residual error sum of squares; Adj R-Squared: Adjusted R-squared; Pred R-squared: Predicted R-squared; Adeq Precision: Adequate precision.

Supplemental Information 2 Relative expression level dataset of eight promoters under different inducer treatments

Supplemental Information 3 MIQE checklist

Supplemental Information 4 Comparison of the relative expression levels of WGFS, GWSF, WGVFS and GWVSF promoters in response to TMV induction

WGVFS: a modified promoter with the V-box inserted at the WGFS promoter midstream position; GWVSF: another modified promoter with the V-box inserted at the WGFS promoter midstream position. CK1 indicates the promoter transgenic plants not treated without inducer treatment. Transcriptional expression levels are relative values, calculated as the ratio of the expression level of each promoter to that of the CaMV35S promoter. Numbers on the bars in the figure indicate the averages of transcriptional expression levels, and different letters after the numbers indicate significant differences at the 0.05 level.

Supplemental Information 5 Transcriptional expressive level means of four Gst1-box positions in eight promoters

Each point corresponds to the average of the relative expression levels of a specific promoter and V-box position (including duplicates).

Supplemental Information 6 Transcriptional expressive level means of four W-box positions in eight promoters

Each point corresponds to the average of relative expression levels for a specific promoter and V-box position including replicates.

Supplemental Information 7 Transcriptional expressive level means of four GW positions in eight promoters

Each point corresponds to the average of relative expression levels for a specific promoter and V-box position including replicates.

Supplemental Information 8 The amplification efficiency of gus gene and standard curve

Supplemental Information 9 Total RNA detection in Arabidopsis thaliana

M lane: DL 2000 DNA Marker.

We extend our sincere thanks go to the editors and anonymous reviewers for their constructive feedback and valuable suggestions that have enhanced the quality of this article.

Additional Information and Declarations

Competing Interests

Author Contributions

Data Availability

The authors declare there are no competing interests.

Zhenchi Huang conceived and designed the experiments, analyzed the data, prepared figures and/or tables, authored or reviewed drafts of the article, and approved the final draft.

Peiyi Cai performed the experiments, prepared figures and/or tables, and approved the final draft.

Meishi Huang performed the experiments, prepared figures and/or tables, and approved the final draft.

Jiating Chen performed the experiments, prepared figures and/or tables, and approved the final draft.

Weixin Gan performed the experiments, prepared figures and/or tables, and approved the final draft.

Limin Wu performed the experiments, prepared figures and/or tables, and approved the final draft.

Xiaoming Li performed the experiments, analyzed the data, prepared figures and/or tables, and approved the final draft.

Zhihua Wu conceived and designed the experiments, analyzed the data, prepared figures and/or tables, authored or reviewed drafts of the article, and approved the final draft.

The following information was supplied regarding data availability:

The data is available in the Supplemental Files.

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
