# Peer review of "Impact of V-box insertion on promoter activity and virus-inducibility in transgenic Arabidopsis thaliana"

_PeerJ, doi:10.7717/peerj.20178_

## Round 0.1 · original submission · Major Revisions

As pointed out in the review, there are certain flaws in data analysis and interpretation. These can be rectified by carefully addressing the queries raised by both reviewers. You are kindly requested to address all the comments thoroughly.

**Language Note:** The review process has identified that the English language must be improved. PeerJ can provide language editing services - please contact us at [email protected] for pricing (be sure to provide your manuscript number and title). Alternatively, you should make your own arrangements to improve the language quality and provide details in your response letter. – PeerJ Staff

Reviewer 1 ·

Basic reporting

The manuscript is clearly organized and presents the research objectives and results logically. The background is well developed, providing adequate context on synthetic pathogen-inducible promoters (SPIPs) and the challenges with natural promoters.

1. Figures and tables are relevant but would benefit from more detailed captions, especially for Figures 4 and 5, to explain experimental conditions and abbreviations.

2. Some minor typographical errors should be corrected.
- Line 436: “ompared” → “Compared”

Experimental design

The research question is clearly defined and addresses a significant gap in the development of SPIPs for improved disease resistance. Experimental procedures are well described, particularly for the construction of synthetic promoters and their evaluation via GUS staining and qPCR. Proper controls (wild-type and CaMV35S promoters) were appropriately used.

1. The manuscript does not clearly state the number of biological and technical replicates used for GUS staining and qPCR analyses. Please clarify how many independent transgenic lines and how many plants per line were used.

2. While statistical analyses using ANOVA are described, it is unclear if post hoc multiple comparison tests (e.g., Tukey’s HSD) were applied to determine significant differences between groups. This should be explicitly mentioned.

Validity of the findings

The conclusions drawn are generally supported by the presented data. The results from both qualitative (GUS staining) and quantitative (qPCR) methods are consistent and validate the observed promoter activities. The impact of V-box insertion position on promoter activity is convincingly demonstrated with statistical support.

1. The manuscript claims that modified promoters retain responsiveness to multiple inducers, but pathogen response was only evaluated with TMV. For broader claims about pathogen-inducible promoters, additional testing with other pathogens (e.g., bacterial or fungal) would be valuable, or at least discussed as a limitation.

2. While GUS staining was performed after low temperature, high temperature, and hyperosmotic stress treatments, no quantitative RT-PCR data were provided. Since the authors claim that the modified promoters “exhibit a broad range of pathogen-inducible activity while remaining unresponsive to non-pathogen factors,” it would strengthen the claim to provide qPCR data for these abiotic treatments. This would confirm whether transcriptional induction is indeed specific to pathogen-related stimuli.

3. The upregulation of the GUS gene in response to TMV, especially in the WGVFS promoter, is modest (less than a 2-fold increase compared to basal expression). Given that the basal activity is already elevated in these constructs, the reliability of TMV inducibility is questionable. A stronger case for inducibility could have been made by testing the effect of varying V-box copy numbers to see if higher numbers of V-box elements result in stronger induction.

4. The modeling results (Figure 7) are briefly mentioned but lack sufficient explanation. Please provide more details on how the theoretical maximum expression values were calculated.

Reviewer 2 ·

Basic reporting

The writing in this manuscript is unprofessional in many places and requires substantial revision.
1. Lines 223–224, 303–304: Transitional phrases are unnecessary when reporting scientific results. The authors should directly present the findings with the appropriate figure numbers in parentheses.
2. Terminology inconsistencies and misuse: In the Introduction, terms such as “synthetic promoter,” “artificial promoter,” and “improved promoter” are used interchangeably. It is more standard to use “synthetic promoter.” Similarly, “minimal” and “minimum” 35S promoter are confused—“minimal promoter” is the correct term. Phrases like “transcriptional expression level” or “transcriptional expressive level” are not scientifically accurate; use “gene expression level” or “transcriptional activity” instead. The term “binary primary function” (line 408) is unclear and should be clarified or removed. In Figure 5, using “times” as a y-axis unit is inappropriate; relative expression should not have a unit since it's a ratio.
3. Results misplaced in Discussion: Several findings are described for the first time in the Discussion section and should be moved to the Results. For instance, lines 369–372 and 402–410 introduce data not previously discussed.
4. Figure legends lack critical information: For Figure 2, clarify whether the different plants shown represent biological replicates of the same transgenic line or independent transformation events. For Figures 6 and S1–S3, explain what each data point represents and the meaning of color coding. For Figure 4, describe the statistical analyses used.
5. Logical and syntactic clarity: Some sentences are confusing and require rephrasing to improve logical coherence and readability. Please check the whole manuscript for this point.
6. Figure 1: It only shows two promoters, yet four are analyzed in the study. Diagrams for the other two promoters should be included.

Experimental design

no comment

Validity of the findings

1. Figure 2: Images of whole seedlings are insufficient to evaluate background expression levels, especially given the variability in GUS staining across plants and leaves. Quantitative GUS assays using at least four biological replicates are strongly recommended. Also, it appears that young leaves of VWGFS plants have higher background expression than mature leaves. If quantification is performed, specify whether mature or young leaves were analyzed.
2. ANOVA (Lines 287–302): The interaction between factors A × C is missing. Additionally, the statistical approach is described as a “three-factor complete factorial randomized block design,” but it is actually a standard three-way ANOVA. This should be corrected.
3. Figure 5:
o Line 303: Clarify what “theoretical expression levels” mean and what model or assumptions were used to calculate them.
o Line 304: Clarify the y-axis values. The manuscript states WGFS and GWSF have negligible transcript levels by “close to 0,” yet the graph shows negative values—how is that possible?
o Bar plot layout: The current layout makes comparison between treatments difficult. A two-dimensional layout (as in Figure 4) would be more effective.
o Discrepancy with other figures: The expression patterns of some promoters (e.g., WGFS and VWGFS) differ substantially from Figure 4 and need clarification.
4. Lines 309–322: This section mostly compares GWSF and WGFS, but the relevance to the main findings is unclear. The focus should be more on the promoters with V-box insertion. Figure 5 also separates the two promoters into different plots, making comparisons between the two promoters difficult. Some interpretations in this paragraph are incorrect or unsupported:
o Line 313: The claim that WGFS shows “markedly higher expression... particularly when the V-box was inserted at the midstream position” is misleading. The figure shows higher expression only for the midstream position. Also, was this difference statistically significant by the pairwise mean test?
o Lines 315–317: The statement about “greater sensitivity of WGFS to TMV” is not supported by the data. Sensitivity refers to inducibility, which is not higher for WGFS in the figure.
o Line 326: The author did not distinguish the absolute expression level and inducibility correctly. Midstream insertion of V-box leads to higher expression across inducers, but this does not necessarily imply higher inducibility since the baseline is also elevated.
5. Figure 7:
o It is incorrect to describe promoters without V-box as having “0 bp distance to the minimal promoter.”
o Fitting a nonlinear model using only three data points is statistically invalid. Nonlinear models require more data to evaluate stability, residuals, and goodness of fit.
6. Lines 369–372 and Figure 6: The authors argue that V-box positional effects are stronger than those of the other cis-elements. However, Figures S1–S3 show statistically significant positional effects for the other elements as well. Additionally, the statistical approach is flawed:
o Data were pooled across different inducers, violating assumptions of normality.
o Mean values are not informative for non-normal distributions, so results should be presented using medians, not means.
o A non-parametric method, such as the Wilcoxon signed-rank test, should be used

Additional comments

The manuscript presents a study on the positional effects of a virus-responsive cis-element, the V-box, in transgenic Arabidopsis plants. While the study yields some interesting and potentially robust findings, it is hindered by significant flaws in data analysis and interpretation (as detailed above). These issues must be carefully addressed, and the manuscript's writing should be revised to improve logical flow and ensure a clear and unambiguous presentation of results.

---

## Round 0.2 · accepted · Accept

The review assessment confirms that the majority of the Reviewers’ concerns have been adequately addressed, including enhanced methodological transparency, and improved narrative coherence.

Reviewer 1 ·

Basic reporting

The authors have responded thoroughly to the reviewer’s comments and substantially improved the clarity of figure captions, particularly for Figures 4 and 5. The revised captions now provide sufficient detail on experimental conditions, treatments, and abbreviations, which will be helpful for readers not familiar with the system. Typographical corrections have also been addressed, and the manuscript appears carefully proofread. Overall, the manuscript is now clearly organized and accessible.

Experimental design

The authors have clarified important methodological details, including the number of biological and technical replicates for GUS staining and qPCR. This addition strengthens the reproducibility and transparency of the study. The clarification on statistical analysis is also appropriate; explicitly noting the use of Holm-Bonferroni post hoc testing adds rigor. The experimental design remains sound and appropriate for addressing the stated research objectives.

Validity of the findings

The authors have acknowledged and addressed limitations raised in the initial review. They now discuss the need to test promoter responsiveness with bacterial and fungal pathogens, as well as the absence of qPCR validation under abiotic stresses. These acknowledgments improve the balance and transparency of the conclusions.
The discussion on modest TMV inducibility and the potential influence of V-box copy number is particularly valuable, as it highlights future directions to strengthen promoter design. The expanded explanation of the modeling results in Figure 7 also resolves the earlier lack of clarity, providing a clear rationale for how theoretical maximum expression values were derived. Together, these revisions strengthen confidence in the validity of the findings while appropriately recognizing limitations.